# Supporting adult unpaid carers via an online dancing intervention: A feasibility/ acceptability study

Joanna Horne[1]*, Louisa Donald[2], Rosaria Gracia[3], Nichola Kentzer[4], Yannis Pappas[5], Mike Trott[6], Jitka Vseteckova[7]

1 School of Psychology and Counselling, The Open University, Milton Keynes, Buckinghamshire, United Kingdom, 2 School of Psychology, University of Bedfordshire, Luton, Bedfordshire, United Kingdom, 3 School of Social Sciences and Global Studies, The Open University, Milton Keynes, Buckinghamshire, United Kingdom, 4 School of Education, Childhood, Youth and Sport, The Open University, Milton Keynes, Buckinghamshire, United Kingdom, 5 Institute for Health Research, University of Bedfordshire, Luton, Bedfordshire, United Kingdom, 6 Centre for Public Health, Queens University Belfast, Belfast, Northern Ireland, United Kingdom, 7 School of Health, Wellbeing and Social Care, The Open University, Milton Keynes, Buckinghamshire, United Kingdom

☯ These authors contributed equally to this work.
* Jo.Horne@open.ac.uk

**Data Availability Statement:** Excerpts of the transcripts relevant to the study are included within the paper.

## Abstract

Unpaid carers often experience poor mental and physical health linked to their caring role. Engagement in physical activity has been shown to alleviate these negative health outcomes, but it is harder for carers to find the time, energy and space to exercise. This qualitative study, based in the UK, explored the feasibility of an online, dance-based physical activity intervention with six female unpaid carers. Five themes resulted from the thematic analysis of the pre- and post-intervention interviews: Perceived physical health benefits of the intervention; Perceived mental health benefits of the intervention; Satisfactoriness of the dance classes; Impact of caring responsibilities on participation; and Suggestions for future classes. Further research is required to measure the effectiveness of the dance intervention in improving mental and physical wellbeing with larger samples including a wider mix of carers in terms of gender, age and health conditions of the care recipients, as well as international samples. Future research should also consider the barriers that some carers may face when accessing an online intervention, and alternative forms of exercise that may appeal to other groups of carers (e.g., male carers, older adult carers).

## 1. Introduction

It is estimated that there are 10.6 million unpaid carers in the UK [1], with approximately 1.5 million thought to be caring for someone with mental health difficulties [2]. Research shows that carers often experience loneliness, social isolation, psychological distress and physical health difficulties linked to their caring role [3–7]. Indeed, one UK-based study found that 21% of carers described their physical health as bad or very bad and 30% of carers described their mental health as bad or very bad [1]. Research has found that unpaid carers often

**Funding:** This work was supported by The Open University Research Enterprise and Scholarship Pump Priming Engagement project fund (to JV). The funders had no role in study design, data collection and analysis, decision to publish, or preparation of the manuscript.

**Competing interests:** The authors have declared that no competing interests exist.

experience sleep disturbance, fatigue, pain, loss of strength, loss of appetite, and weight loss [8, 9]. In relation to mental health, unpaid carers have been reported to experience anxiety, depression, and stress [8–11]. Importantly, carers are known to focus less on their own health needs and report more negative health behaviours than non-carers [12]. Moreover, two thirds of unpaid carers surveyed in the UK state that they have focussed on the care needs of the person they care for, rather than their own needs [13].

Regular and sustained participation in physical activity is beneficial for almost all facets of health including, for example, reduced risk of cardiovascular disease, certain cancers, stress and depression, and improved mental / cognitive health, wellbeing and sleep [14, 15]. Therefore, the potential of physical activity to enhance carers' health is recognised [16], and has various psychosocial benefits [17]. However, prior to the outbreak of COVID-19, 81% of 8,000 unpaid carers in the UK stated that they were not able to take part in as much in physical activity as they would like due to their caring duties [18]. More recent UK research found that 65% of carers had not undertaken vigorous activity in the previous week, and 35% had not undertaken moderate activity, with 51% of carers being classed as inactive (compared to 27% in the general population) [1]. Indeed, although a recent international review found mixed results with regard to differences between carers and non-carers in overall physical activity (which includes physical activity undertaken as part of the caring role), six studies focusing on leisure time physical activity (i.e. sport, exercise and physically active hobbies, rather than physical activity undertaken as part of one's occupation or caring duties) reported that carers engage in less leisure time physical activity than non-carers [19]. The authors of that review argue that whilst leisure time physical activity is beneficial to health, the physical activity involved in the caring role (e.g., cleaning, shopping, physical help, personal care) may be detrimental to health. Barriers to physical activity, identified amongst UK carers, include perception of aging (not wanting to push themselves physically), not wanting to leave the care recipient alone, the care recipient being unable to take part in activities, health conditions, fatigue, lack of time, and difficulties in changing routine for the care recipient [20].

Research by Carers UK suggests that the pandemic, along with the closure of health and social care services in the UK, prolonged periods of lockdowns (national or regional stay at home orders introduced by the UK government) and self-isolation (the requirement for individuals with Covid to avoid contact with others; or the recommendation for vulnerable people, or those caring for them, to avoid contact with others), made it even harder for carers to get any respite [21]. This increased demand resulted in less time available for exercise and carers, who were at increased risk of social isolation before, found themselves further isolated during the pandemic [21]. Furthermore, while group-based exercise programmes have been found to be effective in motivating inactive adults to take part in exercise [22, 23], in-person classes were arguably impossible during the height of the COVID-19 pandemic in the UK. In line with this, there is evidence to suggest that home-based training increased during lockdown [24], and home-based training, including online classes, is recommended to maintain or increase physical activity during any future lockdowns [25]. Internet-based interventions for carers have been shown to lead to positive outcomes [26]. Additionally, research suggests that online exercise classes for older adults (aged 55–89) during the pandemic were convenient, encouraged regular exercise and would be suitable for carers with limited opportunity for exercise outside of the home, but that there were some technological barriers [27].

Online dance interventions may provide carers with easy access to activities that can promote their physical and mental wellbeing, yet the feasibility and acceptability of such interventions for unpaid carers has never been assessed. Therefore, the aim of this study was to assess the feasibility and acceptability of an online dance class with unpaid carers. The intervention was developed during the third UK COVID-19 lockdown in the Spring of 2021 (which

instigated a stay at home order across England), but it can also be offered to carers in normal times, given carers' known barriers to physical activity (e.g., getting respite care, inability to be away from the person they care for) and the relative inability to leave their homes.

## 2. Method

### 2.1 Materials

Semi-structured interviews formed the basis of the data collection at two time points–pre-intervention and post-intervention. Semi-structured interviews were used, as they allow for flexibility in terms of the topics covered by the interviewee, but with the structure of the topic guide to keep the interview on track. Additionally, they enable in-depth conversation, with interviewers able to ask follow-up questions in response to verbal and non-verbal reactions from the interviewee [28], providing elaboration or clarification. The interviews were conducted online, via Zoom, due to Covid-19 restrictions at the time. Online interviews can be problematic due to the difficulties in building rapport, not being aware of distress or disengagement of the interviewee and technological challenges [29]. To mitigate these difficulties, cameras were kept on during the interviews to allow for the non-verbal responses of all parties to be seen, and the interviews were conducted by an experienced and skilled interviewer.

The pre-intervention interview topic guide was designed to gauge the following: physical and mental wellbeing; experiences as carers; expectations for the dance classes; and how the dance classes might fit in with their lives as carers.

The questions on the post-intervention interview were to follow up on topics explored prior to the intervention, with an additional focus on the participants' experiences and perceptions of the dance classes once completed.

### 2.2 Participants

The participants were all female, with ages ranging from 41–61 years (mean = 54). The mean number of years spent as a carer was eight years, with a range of 1–18 years. All six participants cared for family members, including husband ($n = 1$), sons ($n = 2$), daughters ($n = 2$), and niece ($n = 1$), with mental health needs, although for two of the cared for individuals this was in combination with physical health needs. The intensity of their care needs ranged from two (being mild-moderate) to five (intense), with a mean rating of 3.5. Regarding employment, one carer was in paid work full-time, two part-time, and three were full-time (unpaid) carers.

The participants were already reasonably active, with four of the six participants describing regular engagement in activities, including walking, exercise classes, Pilates, ballet, and swimming. Therefore, a general interest in exercise may have been a motivating factor in their willingness to take part, which is something to take into consideration when assessing the feasibility of an exercise intervention for carers. Furthermore, half of the participants had already taken part in the dance instructor's (RG) similar dance classes as part of the support offered through the carers centre from which they were recruited.

### 2.3 Procedure

The recruitment for the intervention was carried out in conjunction with the Carers Centre for Brighton and Hove, which provides advice, activities, and support for unpaid carers of all ages based in Brighton and Hove in the south of England, and is one of a number of carers centres in the UK that have worked with members of the research team on various projects. The Carers Centre emailed the adult carers (aged 18 plus) that used their services with the full information sheet for the study. Carers who were interested in taking part were asked for their

consent for their email address to be passed on to the research team. Those carers were then sent a link to the Physical Activity Readiness Questionnaire (PAR-Q) [30] by the research team, as a means of ensuring that they were physically able to commit to the intervention. The participants who met the PAR-Q eligibility criteria were sent a link to the online consent form and questionnaire.

When participants had completed the online questionnaire, they were invited to take part in a 20-minute pre-intervention interview online. The semi-structured interviews were all led by one member of the research team. The dance instructor (another member of the research team) was also present at the pre-intervention interviews so that participants could ask questions about the dance classes, mention any physical issues that might impact their participation and make requests for specific music to be included in the sessions.

The dance classes took place once a week for six weeks and lasted for one hour. These classes were conducted online, and the participants had the option of joining the live session and/ or playing a recording. Generally, the participants attended the live sessions–there were just two occasions when two participants were unable to attend. Participants were able to access the recordings whether or not they attended the session, and no record was kept of the number of times the recordings were accessed. The programme was designed by one of the research team, an experienced dance instructor, and piloted with groups of carers and non-carers. The dance classes were designed to be enjoyable and easy-to-follow, incorporating dances that focus on conditioning and somatic movement, which encourage the body to move in different ways. The dance classes were split into two main sections.

Part 1 was carried out in a seated position and started with gentle movements exploring the separation of the upper part and lower part of the body, which worked on the spirals of the body, on the breathing and on making space between the torso and the limbs. The session then evolved from spinal movements to lengthening the torso and thinking about the relationship between the chest and pelvis. The movement of the limbs is also integrated with the principle that most of the movement starts from the torso, so that participants could bring awareness to sections such as the lower back and shoulders, but also be receptive to any issues they may have with the integration of the movement of their limbs. The final section of Part 1 was dedicated to the legs, including hamstring stretches, lengthening of the central line, working with the abductors and adductors, hip/knee mobilisation, rotation of the knees, and bringing awareness between the knee and the feet.

Part 2 of the dance class involved 'follow-the-leader' dances, separated into seven songs, incorporating dances from around the world and from different eras. The first song was always danced in a seated position, to facilitate a good transition between the first and the second part of the class. With this format, the participants could also isolate and work with different parts of the body differently to how they would do standing up. Five more songs were used to explore different parts of the body suggested by the rhythm and dance style, such as: hips and isolations with Latin styles; upper body movement and coordination with footwork with Bollywood music; grounded body positioning with African dance; upbeat music working with the whole body with Charleston and Jive. There was always one song before the cool down which was free style, where participants could slow down or have a last moment of high energy. The cool down worked with the whole body in a fluid manner so there was an easy transition from the previous song.

When the intervention was over, participants completed the post-intervention semi-structured interview with the same researcher who conducted the pre-intervention interviews. Note that one participant was unable to take part in the post-intervention interview. The participants were then emailed a debrief form, explaining how their participation contributed to the study, and what to expect concerning the dissemination of the findings.

## 2.4 Data analysis

The focus of this qualitative study was on the views and experiences of the participants, centred around the feasibility of the intervention within their roles as carers. In consideration of this being an open and subjective topic, the theoretical freedom of thematic analysis made it an appropriate fit for the study. Braun and Clarke's model of Reflexive Thematic Analysis [31] was adopted to enable the coding process to be organic and flexible and to take account of the effect of the backgrounds and assumptions of the researchers in the interpretation of the data.

For both the pre- and post-intervention interviews, the process of analysis started with the interviews being transcribed verbatim by a member of the research team. Once the transcriptions were complete, one researcher reviewed all the transcripts and coded the data. Following the coding, the codes were then organised into code categories for each interview, grouping together concepts. These code concepts were then organised into subthemes and themes for each interview. Finally, the themes from all the interviews were mapped as a means of gauging the overarching themes across all interviews, supported by NVivo 11 software [32]. Finally, a discussion took place amongst five members of the research team concerning how to organise the presenting themes and subthemes to best reflect the data.

It is acknowledged that the interpretation of the data is impacted by the subjective positions of the researchers conducting the analysis, resulting from their own experiences and bias. The dance instructor leading the intervention (RG) is also a lecturer and researcher, and a member of the research team involved in this project. The other members of the research team have academic research backgrounds in sport and exercise psychology (NK, MT, JH), health psychology (LD) or public health policy and promotion (JV, YP). It is probable that, individually and collectively, the researchers' position of promoting physical activity to improve public health will have impacted on the interpretation of the findings, particularly regarding the perceived positive impact on mental and physical health.

## 2.5 Ethical considerations

Ethical approval was provided by the Open University Human Research Ethics Committee (HREC/3817/Vseteckova), as well as the Institute for Health Research Ethics Committee (IHREC) at the University of Bedfordshire (IHREC961). With regards to the specific ethical considerations relative to this exercise-based intervention, the physical safety and wellbeing of the participants was paramount throughout. Participants were not offered any incentive to take part in the study, other than the opportunity to attend the dance sessions at no cost. The information sent out to potential participants included the email addresses of two members of the research team so that individuals could ask questions prior to consenting to take part, or at any other time during the study.

All participants were pre-screened for conditions that could be made worse by their participation. Additionally, the dance instructor (RG), who is fully insured and qualified, reiterated to the participants during their pre-intervention interviews that it was their responsibility to take the dance classes at their own pace, and to make appropriate decisions regarding their continued participation in each session if they were feeling unwell or in pain at any point. Each participant's contact and emergency contact details were available to the dance instructor and research assistant, in case something happened whilst the dance classes were taking place.

The semi-structured interviews took place online, via Zoom, at a time that was suitable for the participant with regard to their caring duties, employment and other responsibilities. They were conducted by an experienced interviewer alongside the dance instructor. The participants took part in the privacy and comfort of their own home, without their care recipient present. The interviews were recorded to allow for verbatim transcriptions. Interview transcripts were anonymised prior to secure storage, and original recordings deleted following transcription.

## 3. Results

Five themes resulted from the thematic analysis of the pre- and post-intervention interviews: Perceived physical health benefits of the intervention; Perceived mental health benefits of the intervention; Satisfactoriness of the dance classes; Impact of caring responsibilities on participation; and Suggestions for future classes.

### 3.1 Perceived physical health benefits of the intervention

Prior to the intervention, participants were aware of the potential physical health benefits that the intervention might bring. Weight management was mentioned by two participants as a physical benefit of exercise, as one expressed that they would have gained weight during the COVID-19 lockdown period if they had not been carrying out regular exercise, and another that it has helped them to lose weight:

> "I think I probably would have put on a lot more weight if I hadn't been doing, doing those activities"—Participant 1.

> "so that's good and it's helped me lose weight, you know, just get to a better weight than I wanted to be"–Participant 5.

The latter participant also expressed a motivation to remain fit and healthy as they age, aware that they can enhance their physical wellbeing through a strong and able body. Another participant mentioned improvements in strength and flexibility as a result of exercise:

> "I guess there's a sense of accomplishment at the end of it, erm, if, if you can keep up exercise over a few sessions then, then you do start to see quite quick, I do see, start to see, quickly seeing improvements in my strength and flexibility, or the way my body feels with things like that, so that's all good mental health stuff"—Participant 6.

All six participants stated that they expected to feel some physical benefits from taking part in the exercise-based intervention, which encompassed comments about wanting to experience enhanced flexibility, conditioning, energy, stability, suppleness, and weight loss:

> "I'm hoping that the dancing is gonna give me a bit more energy, get me maybe a bit fitter . . . a bit more flexibility I suppose, less achy, maybe a bit more kind of, yeah, just all of that"–Participant 6.

### 3.2 Perceived mental health benefits of the intervention

Prior to the intervention, the participants highlighted the benefit that physical activity has had, and can have, on their mental wellbeing, which occurred as a theme across all six interviews. This included a boost in mood, overcoming and avoiding feelings of depression, creating more time for themselves, enhancing readiness for the day, and overcoming feelings of pressure:

> "it just really makes me smile, so I, yeah it definitely has. I think if I hadn't have done that I would have been quite depressed"–Participant 1.

Reflecting on the specific benefits of dance-based exercise, one participant described how their ballet class would allow them to 'switch off' from the pressures of their stressful role as a carer, seemingly enhancing their mental wellbeing:

*"last year was really stressful here at home because of my sons and, erm, I was able, especially that hour, to switch off because with the ballet you've really got to listen to the teacher and think with your muscles and everything and really switch, switch off to the problems I had"–Participant 5.*

Somewhat similarly, another participant explained how exercise helped boost their mental wellbeing at a time when they were feeling particularly low and, although they did not specify why they were feeling this way, it could be interpreted that this dip in mood was linked to their caring role, considering the context of the interview topic and direction. Furthermore, while they do not specify which type of exercise video they were watching (dance or not), the following quote again demonstrates the positive impact that an exercise routine had over their mental wellbeing, especially when they were already experiencing a low mood:

*"I started doing that at a time, the little morning exercises, at a time when I was really, really low, erm, and, yeah, I found it really, really, you know some days almost hard to sort of, you know, just face the day really, erm, and it was only, I think it was a friend of mine, just sent, sent me this clip, it was a Youtube clip and she said "oh, well I do this and it helps" and it really does and I don't know why it does and, you know, if I, if I do it I just feel much better prepared for the day"–Participant 2.*

Three participants expressed the hope that by taking part in the dance class intervention, they'd be increasing their 'me time'. Although the participants did not specify what they meant by 'me time', it is assumed that they meant time away from their role as carer, and other responsibilities, where they could re-centre and focus on themselves, thus enhancing their mental wellbeing:

*"just relax, feel relaxed and just have time to myself"–Participant 4.*

*"to feel more relaxed, erm, it gives me more time, a focused time for myself to do something I really enjoy"–Participant 5.*

*"also mentally I think for me is really important to have that time"–Participant 1.*

Linked to this need for 'me time', with two of the previous quotes also mentioning relaxation, it was stated by five participants that they expected to experience a boost in mood, encompassing comments about wanting to feel happiness, enjoyment, relaxation, fun, motivation and increased confidence as a result of the intervention. Again, these expectations are linked to mental wellbeing, demonstrating the importance this has for carers, particularly the following quote, where the participant expresses a desire to be 'free':

*"just sort of really be 'free' and sort of, it sets, it sets me up if it's, you know, it sets me up to feel good for the rest of the day, or it will be something for me to hold onto and remember"–Participant 2.*

### 3.3 Satisfactoriness of the dance classes

The participants had overwhelmingly positive things to say about the content and structure of the dance classes, encompassing the music, dance styles, incorporated exercises, and dance instructor's style.

Firstly, there was a vastly positive response to the music choice, across the board. The mixture of music genres was considered a particularly enjoyable aspect, which had a motivating and energising effect, as demonstrated by the second quote below:

*"I really liked the choice of music, I liked the sort of, the, yeah, I liked the, erm, the, the sort of like the global aspect of it I thought that was really good"–Participant 2.*

*"the music starts, it's something about the music, especially the stuff she plays, you just get up and before you know it you've done the whole hour, which really is brilliant, so it's not an effort at all, it's like in spite of yourself sort of thing"–Participant 3.*

Linked to this, the participants were largely satisfied with the incorporated dances and exercises, and general format of the dance classes:

*"I think the variety of different exercises, erm, so there wasn't particularly a sort of favourite style, I think it was great that there were so many different styles . . . it was a good variety of, of dance styles"–Participant 2.*

There were varied responses regarding the level of difficulty, whereby two participants found the classes to be matched to their abilities, whilst three found them challenging. Areas of challenge were linked to the pace and complexity of the dance moves, as articulated in the following quotes:

*"I mean a lot of them I had quite difficulty keeping up to be honest . . . sometimes I thought 'oh my god this is a bit complicated for me', erm, some of the dance steps and things"–Participant 2.*

*"I found it was a bit fast at times, erm, and it took me a while to realise that I didn't have to keep up all the time"–Participant 3.*

*"I mean I'm not very coordinated, coordinated really, I sort of found, sometimes I found that some of the movements, I had to do two at the same time, I found [laughs], I found that sort of quite hard . . . sometimes it got a bit too fast and I sort of lost it"–Participant 4.*

It was, however, mentioned by three participants (including the latter two) that the classes were designed in such a way that they could go at their own pace, or take a step back if they needed to. This was recognised by those who felt they were able to keep up with the dances (participant 1) and those who struggled at times to follow the routines (participant 3):

*"it was a nice sort of gentle group as well but the routines were such that you could kind of throw yourself into it as much as you wanted, so that was really positive"–Participant 1.*

*"just the fact that I could just sit and watch it if I wasn't, and I thought 'well I'll enjoy the music', so no it didn't really because I didn't feel I had to be 100%, you know, but just by turning it on I found I was doing it"–Participant 3.*

Linked to the latter point, the participants were highly complementary about the dance instructor, who they considered to be supportive and encouraging, whatever their abilities. This is something to take into consideration when contemplating the feasibility of this intervention, when factoring in that there was only one dance instructor (whose individual style was liked by these participants) and that the participants were already familiar with her from the pre-intervention interview. Positive comments were mentioned by all the interviewed participants, as demonstrated by the following quotes:

*"[dance instructor] is very friendly as well and very encouraging so it was just, it was getting that encouragement as well . . . [dance instructor] is very experienced and encouraging"–Participant 1.*

*"[dance instructor] encourages me . . . [dance instructor] was great at reminding me about things but I think to be reminded that, you know, if you do it slower and you're just seated or whatever that's fine"–Participant 3.*

*"I just enjoyed the host [dance instructor] as well, she was really nice . . . I mean she's doing a good job"–Participant 4.*

All the interviewed participants expressed that they were happy with the online delivery and availability of recordings, which was useful for the one participant who experienced technological issues, and another who wasn't able to attend every 'live' session:

*"I had to miss a few classes, so I always did the catch up videos, which actually I felt was just as beneficial, erm, [be]cause it was almost like doing the live classes in some respects anyway . . . but it was good that I knew I could do the catch up"–Participant 1.*

Other benefits of the classes being online included being able to 'let loose' and not worry about other people's judgements (although it was stated by the participants that the general atmosphere was relaxed and unjudgmental), as described by two participants below:

*"I think it's my own personal space at home, I like being at home [be]cause I didn't know the area, so worried about talking to anyone or anything"–Participant 3.*

*"I could dance, I could relax that way because . . . I can put my camera on quite happily but if, if it's sort of like, I mean it's alright one to one but if it's in like a big group I'm quite self-conscious"–Participant 4.*

### 3.4 Impact of caring responsibilities on participation

The participants made references throughout their interviews which indicated the type of care they were responsible for, all of which were mental health needs. It could be posited, therefore, that this type of care revolves around being there emotionally for the cared for person, as articulated by the following participant:

*"my [cared for person's] got mental health problems, for him it's keeping the time when to chat, when to listen, support and that could be any time during the day, you know, it's picking the right time"–Participant 5.*

This provides context for understanding how the type of care required by the cared for persons might have influenced the carers' participation in the intervention, in terms of their availability and the demands placed on them. As such, it is acknowledged that the feasibility of an intervention for carers who are dealing with mental health needs is likely to be different from those who are solely responsible for physical health needs. For example, one participant explained that their cared for person is often asleep during the day, meaning that, although they felt guilt about not getting on with 'jobs' around the home, they had the time and space to engage in the class. This time and space factor was also mentioned by several other participants as a means of allowing their participation:

*"while [cared for person] is asleep generally I'm doing jobs and things because I'm always aware she's gonna get up and she'll be quite low and stuff like that"–Participant 3.*

Linked to this, aside from work commitments (mentioned by a few participants) and caring for a toddler (mentioned by one), the carers largely took the view that their caring role didn't stop their participation. Nevertheless, whilst the majority of the participants did not have physical demands per se, the mental health needs of the samples' cared for persons were enduring, and the impact that caring had on the carers' energy levels, availability and readiness for the dance classes each week were still affected by their caring role. For example, three participants explained that, while they looked forward to the classes each week, immediately before they sometimes felt they lacked the energy/time (which soon came when they started the class):

*"most weeks I felt too tired to be honest, like 'what am I doing? I can't do it this week' and, erm, I've got too much else to do and before I would have not, you know, I'd have thought 'oh no I've got this to do'"–Participant 3.*

*"I think sometimes you get sort of days where you've felt you haven't been good enough or done something and you feel bad about it, that type of thing and then it makes you feel tired and stressed out over things"–Participant 4.*

One participant also mentioned that appointments related to their cared for person didn't, but could have, affected their participation, whilst another similarly explained that their cared for person frequently contacts them throughout the day, relying on their support, which might also have, but didn't, affect their participation:

*'I suppose appointments and things like that, if they came up'–participant 4.*

*"in terms of boundaries as well because although he doesn't live here he's sort of constantly calling"–Participant 2.*

An important element to consider with feasibility was the presence of the cared for person whilst the dance classes were taking place, and whether this had an impact on the carer's engagement and enjoyment. The participants did mention having the space to take part, despite the fact that all but one of the participants were living with their cared for person. There were mixed feelings about their desire to have their cared for person 'join in'/be present whilst the dance classes were taking place, where one did not mind, and one wanted this time solely for themselves, as described below:

*"sometimes he did come around in, in the background, there was one point actually that, that he did walk in and I was, I was doing some shimmying or something, sort of walked in and walked out again . . . so he did kind of wander in on occasions . . . I'd love it if my husband would take part in the dancing"–Participant 1.*

*"I always would rather do things away from, away from them"–Participant 2.*

Other participants simply mentioned that their cared for person was somewhere else in the house/outside whilst the dance classes were happening, and either had no interest or were occupied whilst they were taking place, and so did not try and join in.

### 3.5 Suggestions for future classes

In terms of the suggestions for improvement/enhancements, there were only minor comments which focused on additions, as opposed to changes, to the dance class structure and format. This included wanting to have access to the playlist, whereby one participant highlighted that

they enjoyed the music and would have liked to know which songs were used so they could practice in their own time:

> "I would have loved to have known who the music was [be]cause then you can play the list and, and go through it yourself"–Participant 3.

In line with this, other additions to the classes included wanting to be given an instructional break-down of the dances, or more technical information about the dances, to again support them with practising/preparing in their own time, and generally having a better idea of the routines. This was mentioned by three of the participants:

> "I suppose my, I don't know if this is appropriate but I suppose my suggestion was, although I really loved the physical bit at the beginning, I think that was quite long and if we could have made that a bit shorter and then maybe had some of the steps, sort of like the technical aspects of the dance and then went on to do the dance I think that would have been more beneficial to me"–Participant 2.

In terms of social interaction, four of the participants mentioned that they would have liked the chance to be able to engage with the other participants in more depth, to see how they were progressing and enjoying the sessions, as well as for general support:

> "the other thing I would have liked is, first thing is, I think it would have been, especially as we were doing it on Zoom, especially because a lot of us, you know, feeling quite isolated, I think it would have been nice to have had a check in right at the beginning, just like 'how are you'"–Participant 2.

> "even if [dance instructor] wasn't there it would have been really nice to have had a little break out chat room or something to ask how we all felt as well, giving each other encouragement"–Participant 3.

Linked to this, even though all the interviewed participants expressed that they were happy with the online format of the dance classes, two stated that they would have liked to meet up with the other participants in a 'real life' setting, even suggesting a mixture of online and face-to-face sessions:

> "I think what would have been really nice would, you know, I would have liked to have, really have been, erm, seeing people in, in the flesh"–Participant 2.

> "online and in class would be like absolutely perfect really because I think my other problem is that I do like going to classes but you can't go every week and then you've lost that consistency and you think 'oh why bother', so I think, I think online for me was fantastic, it was, yeah, was fine, that's the only thing"–Participant 3.

There were two participants who explained that the recorded sessions were extremely useful for when they were unable to attend the 'live' class/they were having technical issues. It is worth considering, however, that only two of the participants couldn't attend the 'live' sessions, on only two occasions, and so engagement was found to be generally good. Nonetheless, these comments highlight the usefulness of being flexible with the delivery of an online intervention, particularly for carers who are often under time constraints. In line with this, suggestions for a 'better' time varied between two participants–indicating that there really is no universally

'ideal' time to carry out a dance class—between having the classes earlier on in the morning (to start the day with), during lunch time, or more towards the evening, as articulated below:

> "I do work part time as well, so for me probably maybe a little bit later, erm, in the day, or maybe more around lunchtime might have been better for me"–Participant 1.

> "I think maybe it could have been a little bit earlier in the morning, so sort of started with that rather than sort of, yeah, having doing things either side of it, I think if it was the first, you know, if it has been like this, half 9, that would have been good"–Participant 2.

## 4. Discussion

The current pilot study aimed to determine the feasibility and acceptability of an online-based dancing intervention for unpaid adult carers. It is worth noting that the participants already had an understanding and awareness of the benefits of exercise, prior to taking part in the intervention and the majority of the participants led fairly active lifestyles. It could be argued that the sample of carers in this study were an already motivated group, for whom an exercise programme was not difficult to engage with. Nevertheless, participants viewed the intervention as encouraging them to make more 'me time', and they reported that the dance classes were very accessible, which may have been the key driving factor to their engagement, rather than their high motivation.

It is interesting to note that the volunteer sample consisted entirely of working-age females, caring primarily for individuals with mental health difficulties, despite the invitation going to a wider group of adult carers. It may be that the dance classes were more attractive to female carers than to male carers. There are known gender differences in preferences for different types of physical activity, with women being more likely than men to prefer activities at a fixed time, supervised, with others of the same sex and age, and being less likely than men to prefer activities that are competitive, vigorous, skilled or outdoors [33]. Additionally, male carers may be less likely than female carers to see a need for a physical activity intervention. Certainly, there is evidence that males find caring less burdensome and stressful than females [34–39], which may be because male carers use more coping strategies, such as problem-solving and treating the role as work, therefore becoming less emotionally involved and less likely to experience distress [35, 40, 41]. It has also been argued that female carers report poorer health than male carers [42, 43]. So, if male carers experience less stress and burden from the caring role and fewer health issues, they may feel less need to gain support by attending exercise classes. However, research has also suggested that men are less likely to self-identify as carers, prefer to remain independent of service support, and may feel guilty about seeking help [44]. Furthermore, females report loneliness more often than males [45], and female carers report higher levels of social deprivation [36], so there may be a greater need for female carers to attend sessions in order to connect with others, especially given the social isolation that is associated with being a carer.

It is also possible that the online aspect of the intervention was a barrier for older adult carers, who may not have internet access or lack online literacy. Despite several studies finding that online physical activity interventions are acceptable, enjoyable and useful to older adults [46–53], some research has identified barriers for this age group in terms of computer / internet access [54]. Nevertheless, evidence from the UK suggests that older adults have become more familiar with, and accepting of, the use of online technology for accessing activities as a result of the restrictions during the COVID-19 pandemic [55]. It is also recognised that as carers get older, they are more likely to feel that they are doing as much exercise as they would like to [56].

The first two themes highlight the participants' beliefs concerning the positive effects of exercise on their mental and physical wellbeing. As indicated in the introduction, research has consistently demonstrated that exercise can be beneficial for carers [16, 17], and it is therefore relevant that the participants were aware of these benefits prior to their engagement with the dance intervention. This highlights that the participants' engagement with exercise and their lived experiences of the benefits may have been a motivating factor in their willingness to take part in the intervention [20, 57], which is a factor to take into consideration when addressing the feasibility of this intervention at a wider level. However, it is also plausible that the ease with which the classes fit into the carers' lives is a driving factor to engagement. Participants expected that the dance classes would boost their mood, increase 'me' time and lead to physical improvements. As outlined within the introduction, online exercise interventions, including dancing classes, have the potential to provide carers with accessible programmes which can enhance their physical and mental wellbeing [14–17].

The third theme identified the participants' positive views of the content and structure of the intervention, with comments focusing on the supportiveness and non-judgemental attitude of the dance instructor, the interesting and global choice of music, and the mix of exercises–which although challenging for some, were appropriate for those of all abilities. This suggests that the intervention was acceptable to carers. The importance of having an instructor who is empathetic and has a clear communication style, alongside a programme that takes account of health conditions and fitness levels, is also highlighted in other research [46].

This final theme highlighted some suggestions for improvements, including wanting more instruction, access to the play list to practice in their own time, and increased opportunity to engage with other participants. This final suggestion is particularly pertinent given the isolation experienced by carers [3–5]. Indeed, there being a social aspect to the activity has been identified as a facilitator to carers engaging with physical activity [57].

Promisingly, the findings indicate that there were no insurmountable barriers to the carers' participation, and their engagement with the dance sessions each week was generally good. However, it is worth acknowledging that this may have also been influenced by the fact that the needs of the participants' cared for persons were largely mental and emotional, as opposed to physical. In line with this, as highlighted by the participants, the online format of the dance intervention was particularly convenient, and the 'catch up' videos were especially useful, further demonstrating the suitability of online-based physical activity programmes for carers who are time-limited.

## 4.1 Limitations

Whilst producing promising findings, there are some limitations to this feasibility study. First, the sample was limited to only six carers, all of whom were female, of working age and based in the UK. Primarily, the participants were caring for individuals with mental health difficulties, although two of the care recipients also had physical health issues. Additionally, the online dance intervention requires participants to be computer / internet literate and to have the physical space to perform the routines in their own homes. Finally, the findings presented here are based only on interview data, without any objective measurement of health and wellbeing status over the period of the study.

Further research is required to establish the effectiveness of the dance intervention using a larger UK sample, which is more representative of the population of carers in terms of gender, age and health conditions of the care recipients, as well as international samples. Future research should also consider the barriers that some carers may face when accessing an online intervention, and investigate alternative forms of exercise to dance, for example yoga, Pilates

and Tai-Chi that may appeal to other groups of carers (e.g., male carers, older adult carers). Pre- and post-intervention measures of mental wellbeing, physical health and activity levels would be useful to demonstrate the effectiveness of the programme.

## 4.2 Conclusion

The findings of this feasibility study demonstrate that an online exercise programme of this kind has the potential to be a feasible and acceptable intervention for carers. Further research, as outlined above, would allow for the expansion of this intervention, working with stakeholders and carers organisations to support the implementation of online dancing sessions, as a provision regularly offered to carers. The suggestions given by participants for improvement of the intervention were only minor and easily adaptable for a future roll-out of this intervention. To conclude, this feasibility study provides promising findings, which can be utilised and built upon in future work which aims to improve the wellbeing of unpaid carers.

## Author Contributions

**Conceptualization:** Joanna Horne, Rosaria Gracia, Nichola Kentzer, Yannis Pappas, Jitka Vseteckova.

**Data curation:** Louisa Donald.

**Formal analysis:** Louisa Donald.

**Funding acquisition:** Nichola Kentzer, Jitka Vseteckova.

**Investigation:** Louisa Donald, Rosaria Gracia, Nichola Kentzer, Mike Trott, Jitka Vseteckova.

**Methodology:** Joanna Horne, Louisa Donald, Rosaria Gracia, Nichola Kentzer, Yannis Pappas, Mike Trott, Jitka Vseteckova.

**Resources:** Rosaria Gracia.

**Supervision:** Yannis Pappas, Jitka Vseteckova.

**Writing – original draft:** Joanna Horne, Louisa Donald.

**Writing – review & editing:** Joanna Horne, Louisa Donald, Rosaria Gracia, Nichola Kentzer, Yannis Pappas, Mike Trott, Jitka Vseteckova.

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
