## [Decision Letter · Decision Letter 0]

28 Jul 2023

PGPH-D-23-00674

Supporting adult unpaid carers via an online dancing intervention: A feasibility/acceptability study

Dear Dr. Horne,

Thank you for submitting your manuscript to PLOS Global Public Health. After careful consideration, we feel that it has merit but does not fully meet PLOS Global Public Health’s publication criteria as it currently stands. Therefore, we invite you to submit a revised version of the manuscript that addresses the points raised during the review process.

We look forward to receiving your revised manuscript.

Kind regards,

Jianhong Zhou

Staff Editor

Journal Requirements:

Additional Editor Comments (if provided):

Reviewers' comments:

Reviewer's Responses to Questions

**Comments to the Author**

1. Does this manuscript meet PLOS Global Public Health’s publication criteria? Is the manuscript technically sound, and do the data support the conclusions? The manuscript must describe methodologically and ethically rigorous research with conclusions that are appropriately drawn based on the data presented.

Reviewer #1: Yes

Reviewer #2: Partly

2. Has the statistical analysis been performed appropriately and rigorously?

Reviewer #1: N/A

Reviewer #2: N/A

3. Have the authors made all data underlying the findings in their manuscript fully available (please refer to the Data Availability Statement at the start of the manuscript PDF file)?

Reviewer #1: Yes

Reviewer #2: Yes

4. Is the manuscript presented in an intelligible fashion and written in standard English?

Reviewer #1: Yes

Reviewer #2: Yes

5. Review Comments to the Author

Reviewer #1: The paper is very well written. It is clear and concise and well referenced.

The introduction is excellent. The authors have set out the context of the physical and mental health problems experienced by unpaid carers and the benefits of, and barriers to, increasing their physical activity. The purpose of the research is clear. This is an interesting and potentially important intervention for unpaid carers.

The methods section is again well written and clear but it is missing a section on research design, looking at the pros and cons of undertaking qualitative semi-structured interviews with a sample of unpaid carers. I agree with the research design but it would have been good to see a justification for it. I would have liked to read something about the intervention itself. What sort of dance? Was there a choice of style? Could the carers take part in the live class and also have access to the pre-recorded video if they wanted to watch it again or do the session again? The section on data analysis is overly brief. I would like to read more about how you analysed the data - was it transcribed and coded manually? Were transcripts (or a sample of them) coded by more than one researcher? How did you analyse the coded data? Also the Braun and Clarke reference is quite old now. There are new materials by Braun and Clarke which reflect their further thinking around thematic analysis. Again, the section on ethical considerations is overly brief - was the care recipient present during the pre and post interviews or was the carer advised/able to be interviewed in private? Were there any incentives for participants to take part (e.g. a shopping voucher as a thank you)? Did potential participants have the opportunity to ask questions ahead of the interviews? Also, did some carers take part in pre-intervention interviews but not post-intervention interviews?

The results are very interesting, though very descriptive. I don't understand how participants were familiar with the dance instructor from the pre-intervention interview - was the interviewer also the dance instructor?

The discussion is brief but good and contextualises the findings within the wider literature. However, I would like to see a limitations sub-section in the discussion - perhaps considering carers who are not computer/internet literate and/or those who do not have the physical space to dance in their own homes? Perhaps before you talk about the expansion of the programme and making online dancing classes a provision regularly offered to carers, it would make sense to first test the intervention to see if/to what extent it actually does improve mental and/or physical wellbeing and also how the intervention works for carers of people with physical care needs or learning disabilities etc.

There is a good range of relevant and recent references and the referencing style is excellent.

Line 31, insert the word 'between' before 21%.

Line 445, suggest remove the words 'in this manuscript'.

Line 501, should 'or' be 'of'?

Line 513, 'o flongitudinal' should be 'of longitudinal'.

Reviewer #2: I really enjoyed reading this interesting paper about an online dancing class as an intervention to increase carers’ physical activity- these sorts of interventions are much needed and this one showed some promise. If the issues shown below are resolved, I think it would be an interesting and useful paper. I have listed some major overall points and then some issues line by line.

1. You need to be clearer that the research us in the UK and describe the UK context. Be clear when you cite UK-based research compared to international research. How could this intervention be used around the world? This is a Global audience. Which populations might it be useful for outside of the UK?

2. You need to be clear when talking about Covid-19 pandemic and the policies that resulted from the pandemic – such as, self-isolation and limited contact. Each country had a unique story and you must be clear about the UK context.

3. There is a potential ethical issue arising from the dance teacher also being an author – you just need to be transparent about their involvement and how you mediated any potential bias.

4. I wasn’t sure how the intervention was developed – this seems an important aspect considering it is a feasibility study

5. Need more information about recruitment/who got the email/ any pre-existing links to Brighton carers centre etc

6. The use of thematic analysis is partly incorrect and is confusing- please see my comments below. Tightening up this part of the paper would improve it a lot. At the moment all the ‘themes’ and ‘subthemes’ are confusing- especially in the abstract. The data is so interesting – but it is masked by the confusing write up. Some of the themes are mislabelled themes- when really they are questions. Think about what themes are – you definitely have some themes but also just have some responses to questions in there- which is fine!

7. Be careful when making assumptions about mental health carers and their caring loads.

8. Needs to connect to the wider literature on a) different types of carers b) online exercise or group interventions and their barriers and facilitators c) who joins these interventions and who might be excluded. This is missing in the opening and the discussion. There is a whole literature out there on online interventions – and how they have been used since the pandemic. And on different types of carers and who engages and who doesn’t – and who does or doesn’t take part in physical activity etc. Need a lot more on this.

9. Reflection on how to broaden engagement and participation to address the clear issues in the heterogeneity of your sample, or some discussion about this.

10. Finally, the whole paper could do with a good edit – there are some bits that don’t make sense or could be a bit better explained. I have tried to point these out as I came across them.

Line by line:

Abstract- needs to state it is the UK.

LINE 14 – not sure about the word ‘suffer’ – experience may be better

Repetitive abstract – ‘revealed two key themes’

Really not keen on the word suffer-try to choose different words.

Line 44- not clear this research is in the UK or what methods used

Line 45- same.

Line 51-52 leisure time physical activity’ -maybe say this better?

Line 54- ‘Barriers to physical activity include increasing aging’ – this is a bit ageist –What do you mean by this exactly? The link between ageing and health/mobility?

Line 56-57 – Again make this clear it is the UK context – be specific it is research by Carers UK etc

Line 56 Could be a new paragraph start, following with line 59 & 60.

Line 59. Think you need to explain the pandemic a little bit more- make it clear the UK context as this is a global health journal.

Line 61 ‘Increased demand on coping’ isn’t very well written

62- ‘It wasn’t impossible to take part in group exercise classe’s- it was actually very common in the pandemic- online. It was actually very common. Be clearer.

65- during future lockdowns?

71- “in non-quarantine periods” is a strange term- you could say, in normal times… etc.. Or link into arguments about how the digitalisation of interventions during Covid-19 pandemic has resulted in a changed landscape for UK service delivery. There are some papers circulating on this topic now- long lasting impacts of Covid-19 pandemic.

76 – do you mean topic guide – not familiar with the word schedule?

90-91 Emailed to who?- How did you gain access to Brighton Carer Service? What is the Brighton Carer Service- what does it do? Explain for a non-uk audience. What types of carers are registered.

97- Who curated this dance activity? How? Was it evidence informed? How long did it take? Was it piloted? More about the intervention please.

108-109 – More information on data analysis is needed here. Who analysed the data? How? Why Braun and Clarke’s model?

117- The dance teacher was a co-author? I feel like more needs to be made of this. I think this raises an ethical question in itself. You need to be more transparent about their exact role in the study, how you mediated for any potential bias etc. The authors prior connection to the Brighton Carers Centre. The dance teacher- did she know any of the participants before the study through the dance community? Etc. Did the authors know any of the participants in the dance classes before the study? It is fine if you do- but that needs to be explained and taken into consideration.

129- The way you’ve talked about this as themes seems incorrect for me – For example, Theme 2 in Table 1- I think there are three themes in response to the question on expectations. But expectations isn’t a theme itself- it is an interview question.

Similar applies across all themes.

Theme 1 could be – in response to why they wanted to take part themes were identified which could be 1 and 2. But that isn’t a theme? It is a subset of questions.

Again- apply to 3.2 – the use of themes isn’t right- a theme isn’t an interview question/topic. It is how you organise the information in the responses to a question/topic.

200- mental well-being side of health, is a clumsy expression.

220. Why did someone drop out? You say 5 here.

272- Individual style particularly likeable- could you say, individual style was liked by these participants?

355- be careful with how you refer to mental health carers and the assumptions you make here about the challenges of being a mental health carer and their availability. Mental health caring can be just as challenging and lead to the same lack of time.

Other comments.

Think about- who didn’t come forward? Who didn’t volunteer?

Need more info about how this research could be generalised/transferred to other contexts etc

I want to know how many of the participants did each class and what method they used? Online/offline

6. PLOS authors have the option to publish the peer review history of their article (what does this mean?). If published, this will include your full peer review and any attached files.

**Do you want your identity to be public for this peer review?** For information about this choice, including consent withdrawal, please see our Privacy Policy.

Reviewer #1: No

Reviewer #2: No

---

## [Decision Letter · Decision Letter 1]

19 Dec 2023

Supporting adult unpaid carers via an online dancing intervention: A feasibility/acceptability study

PGPH-D-23-00674R1

Dear Dr Horne,

We are pleased to inform you that your manuscript 'Supporting adult unpaid carers via an online dancing intervention: A feasibility/acceptability study' has been provisionally accepted for publication in PLOS Global Public Health.

Best regards,

Julia Robinson

Executive Editor

Reviewer Comments (if any, and for reference):

Reviewer's Responses to Questions

**Comments to the Author**

1. If the authors have adequately addressed your comments raised in a previous round of review and you feel that this manuscript is now acceptable for publication, you may indicate that here to bypass the “Comments to the Author” section, enter your conflict of interest statement in the “Confidential to Editor” section, and submit your "Accept" recommendation.

Reviewer #1: All comments have been addressed

2. Does this manuscript meet PLOS Global Public Health’s publication criteria? Is the manuscript technically sound, and do the data support the conclusions? The manuscript must describe methodologically and ethically rigorous research with conclusions that are appropriately drawn based on the data presented.

Reviewer #1: Yes

3. Has the statistical analysis been performed appropriately and rigorously?

Reviewer #1: N/A

4. Have the authors made all data underlying the findings in their manuscript fully available (please refer to the Data Availability Statement at the start of the manuscript PDF file)?

Reviewer #1: Yes

5. Is the manuscript presented in an intelligible fashion and written in standard English?

Reviewer #1: Yes

6. Review Comments to the Author

Reviewer #1: Thank you for revising the manuscript and addressing the reviewer comments. I am happy that you have responded to the reviewer comments fully and carefully. This is a very interesting and well-written paper.

7. PLOS authors have the option to publish the peer review history of their article (what does this mean?). If published, this will include your full peer review and any attached files.

**Do you want your identity to be public for this peer review?** For information about this choice, including consent withdrawal, please see our Privacy Policy.

Reviewer #1: No
